# Absence of Host-Specific Genes in Canine and Human *Staphylococcus pseudintermedius* as Inferred from Comparative Genomics

**DOI:** 10.3390/antibiotics10070854

**Published:** 2021-07-14

**Authors:** Alice Wegener, Els M. Broens, Linda van der Graaf-van Bloois, Aldert L. Zomer, Caroline E. Visser, Jan van Zeijl, Coby van der Meer, Johannes G. Kusters, Alex W. Friedrich, Greetje A. Kampinga, Gregorius J. Sips, Leonard Smeets, Manfred E. J. van Kerckhoven, Arjen J. Timmerman, Jaap A. Wagenaar, Birgitta Duim

**Affiliations:** 1Department of Infectious Diseases and Immunology, Faculty of Veterinary Medicine, Utrecht University, Yalelaan 1, 3584 CL Utrecht, The Netherlands; a.c.h.wegener@uu.nl (A.W.); E.M.Broens@uu.nl (E.M.B.); l.vandergraaf@uu.nl (L.v.d.G.-v.B.); a.l.zomer@uu.nl (A.L.Z.); a.j.timmerman@uu.nl (A.J.T.); J.Wagenaar@uu.nl (J.A.W.); 2Amsterdam UMC Location AMC, Meibergdreef 9, 1105 AZ Amsterdam, The Netherlands; c.e.visser@amsterdamumc.nl; 3IZORE Centre for Infectious Diseases Friesland, 8900 JA Leeuwarden, The Netherlands; j.van.zeijl@izore.nl (J.v.Z.); c.vandermeer@izore.nl (C.v.d.M.); 4Utrecht Medical Center, Heidelberglaan 100, 3584 CX Utrecht, The Netherlands; hans@hanskusters.nl; 5Department of Medical Microbiology, University Medical Center Groningen, University of Groningen, 9700 RB Groningen, The Netherlands; alex.friedrich@umcg.nl (A.W.F.); g.a.kampinga@umcg.nl (G.A.K.); 6Reiner Haga MDC, Reinier de Graafweg 7, 2625 AD Delft, The Netherlands; g.sips@rivm.nl (G.J.S.); l.smeets@reiner-mdc.nl (L.S.); 7Erasmus University Medical Centre (Erasmus MC), Department of Medical Microbiology and Infectious Diseases, 3015 GD Rotterdam, The Netherlands; 8LabMicta, LabMicTA Boerhaavelaan 59, 7555 BB Hengelo, The Netherlands; m.vankerckhoven@labmicta.nl; 9Wageningen Bioveterinary Research, 8200–8249 Houtribweg, 8221 RA Lelystad, The Netherlands

**Keywords:** *S. pseudintermedius*, comparative genomics, host association, antimicrobial resistance

## Abstract

*Staphylococcus pseudintermedius* is an important pathogen in dogs that occasionally causes infections in humans as an opportunistic pathogen of elderly and immunocompromised people. This study compared the genomic relatedness and antimicrobial resistance genes using genome-wide association study (GWAS) to examine host association of canine and human *S. pseudintermedius* isolates. Canine (*n* = 25) and human (*n* = 32) methicillin-susceptible *S. pseudintermedius* (MSSP) isolates showed a high level of genetic diversity with an overrepresentation of clonal complex CC241 in human isolates. This clonal complex was associated with carriage of a plasmid containing a bacteriocin with cytotoxic properties, a CRISPR-cas domain and a pRE25-like mobile element containing five antimicrobial resistance genes. Multi-drug resistance (MDR) was predicted in 13 (41%) of human isolates and 14 (56%) of canine isolates. CC241 represented 54% of predicted MDR isolates from humans and 21% of predicted MDR canine isolates. While it had previously been suggested that certain host-specific genes were present the current GWAS analysis did not identify any genes that were significantly associated with human or canine isolates. In conclusion, this is the first genomic study showing that MSSP is genetically diverse in both hosts and that multidrug resistance is important in dog and human-associated *S. pseudintermedius* isolates.

## 1. Introduction

*Staphylococcus pseudintermedius* is found both as a commensal bacterium as well as an opportunistic pathogen in dogs. *S. pseudintermedius* in dogs is associated with skin, soft tissue and systemic infections similar to *S. aureus* infections in humans. Over the last decades, *S. pseudintermedius* is increasingly recognized as a potential zoonotic pathogen of canine origin in elderly and immunocompromised humans [1]. There is an increase in reports of *S. pseudintermedius* infections in humans, which might be at least partially explained by the implementation of MALDI-TOF MS in routine diagnostics facilitating proper identification of coagulase-positive staphylococci [2,3,4,5].

The epidemiology of *S. pseudintermedius* in human infections is poorly studied. Human infections have been reported to be mainly caused by methicillin-susceptible *S. pseudintermedius* (MSSP). This is often thought to result from transmission of MSSP between dogs and humans within the same household [6,7]. Transmission of methicillin-resistant *S. pseudintermedius* (MRSP) between dogs has been frequently observed, leading to long time carriage with possible re-infections. Dog-to-human transmission was infrequent and no long-term carriage in humans was observed [8]. A single case of human-to-human MRSP-transmission has been described [9].

In contrast with *S. aureus* which has been isolated from multiple host species and shows frequent acquisition, or loss of host-associated genes [10], *S. pseudintermedius* seems to be more host-restricted. However, information on the genetic variation and the mechanisms that allow adaption of MSSP to humans is scarce. A study on *S. pseudintermedius* adherence properties to corneocytes revealed a general preference for canine corneocytes compared to human corneocytes [11] and another study identified a cell-wall-associated protein with high binding strength to canine fibrinogen compared to fibrinogen of other host species [12]. In order to identify potential host-specific genes and clones we performed comparative genomics of methicillin-susceptible *S. pseudintermedius* from canine and human origin.

## 2. Results

### 2.1. MSSP-Infections Determinants

Patient information was incomplete for 7/32 (22%) of the obtained human isolates. Most human MSSP isolates were from wound infections (*n* = 18). Other infections were ear (*n* = 2), joint (*n* = 2), skin (*n* = 2), systemic (*n* = 1), urinary tract (*n* = 1) and rectum infections (*n* = 1). The age of patients from 24 cases varied between 48 and 86 years; 1 patient was 6 years old (Appendix A). Patient isolates were obtained from hospitals located in six different provinces. The canine MSSP isolates were isolated from different body sites. The majority of canine isolates were from ear infections (*n* = 9), followed by skin infections (*n* = 8), wound infections (*n* = 5), urinary tract infections (*n* = 2) and a joint infection (*n* = 1) (Appendix A). All isolates were obtained between 2014 and January 2019. Isolates were isolated from different patients at different time points and are to the best of our knowledge unrelated epidemiologically.

### 2.2. S. pseudintermedius Phylogeny

The genetic relatedness of canine and human MSSP isolates is visualized in a phylogenetic SNP tree of the core genome (size of 2,170,170 bp) in comparison with included publicly available genomes [12,13,14] (Figure 1). This placed our results in a wide epidemiological context and shows that the isolates of this study are dispersed among the high genomic diversity of MSSP isolates.

To zoom in on the genome comparison of the dog and human isolates from this study the genotype details are shown in a phylogenetic tree in Figure 2. The extracted MLST types were superimposed, and the studied isolates belonged to 50 different sequence types (ST). The observed phylogenetic diversity between MSSP genomes was high overall and showed 1 cluster containing 10 genomes. This cluster, corresponding to clonal complex CC241 (comprising of ST241, ST941, ST1379, ST1350, ST1360), dominated, and consisted of seven human isolates (comprising of ST241, ST941, ST1379) and three canine isolates (comprising of ST241, ST1350, ST1360). While CC241 was overrepresented in humans, the difference in proportion of isolates from both hosts present in the CC241 cluster was not statistically significant (*p* = 0.487). Besides the genome cluster belonging to CC241, only two other genetically related pairs were identified, with 1 pair consisting of a canine and a human isolate of ST989 with 55 SNP differences and 1 pair with 2 human isolates of ST985 from the same hospital with 8 SNP differences in their core genome. All other STs were represented by a single isolate (Figure 2).

Figure 2 shows a core-genome SNP tree of canine and humans (bold italic) isolates, presence (filled circles) or absence (empty circles) of antimicrobial resistance genes, presence of mobile genetic element PRE25-like [15] and plasmid P222 [16] (filled triangles), and sequence type (ST). Clones are shown by coloured branches (CC241 in red, ST989 in green, ST985 in purple). The length of the branch represents the number of SNPs.

### 2.3. Antimicrobial Resistance Genes

In *S. pseudintermedius* it has been shown that antimicrobial resistance genotypes can accurately predict phenotypical antimicrobial resistances [14,17,18]. Multidrug resistance (MDR) (i.e., carriage of resistance genes to 3 or more classes of antimicrobials) was identified in 13/32 (41%) human isolates and 14/25 (56%) canine isolates (Appendix A, Figure 1 and Table 1). No resistance genes were identified using Resfinder in 7/32 (22%) human isolates and in 2/25 (8%) canine isolates. Following the statistical analysis for host association, no statistically significant difference was found neither for the prevalence of predicted MDR in human versus canine isolates (*p* = 0.249), nor for isolates without resistance genes (*p* = 0.273).

The distribution of the identified antimicrobial resistance genes in MSSP from both hosts is presented in Table 1. The penicillin resistance gene *blaZ* was found in the majority of isolates from both hosts (48/57) (84%). Aminoglycoside resistance genes *ant6-Ia* and *aph(3′)-III* were found in 14/25 (56%) of the canine isolates compared to 14/32 (44%) in human isolates. The gene *aac(6′)-Ie-aph(2”)-Ia* was detected in two canine isolates. The chloramphenicol resistance gene *cat*_(pC221)_ was present in 13/32 (41%) of human isolates and 9/25 (36%) of canine isolates. The macrolide resistance gene *erm(B)* was present in 13/32 (41%) of human isolates and 10/25 (40%) of canine isolates. The tetracycline resistance gene *tet(M)* was present in 11/25 (44%) of the canine isolates and 6/32 (19%) of the human isolates. The folate pathway inhibitor resistance gene *dfrG* and lincosamide resistance gene *Inu(A)* were not present in human isolates and only in 1/25 (4%) of the canine isolates. No statistically significant host association was found for the differences in resistance gene presence in canine and human isolates.

### 2.4. Host-Associated Genes

We did not identify gene presence or absence significantly associated with either canine or human *S. pseudintermedius* isolates using Roary (Appendix A).

The genes *spsL* and *spsD* were studied specifically using alignments, as they were previously linked with host specificity [11,19]. The complete gene encoding the fibrinogen binding protein *spsD* was detected in 25 human isolates and 21 canine isolates. The fibronectin and fibrinogen binding protein *spsL* was present in 30 human isolates and 24 canine isolates. Alignment to reference genes *spsD* and *spsL* from ED99 showed a high level of sequence diversity across the gene with no clear clustering of variants of either dogs or human isolates (Appendix A).

As CC241 was overrepresented in human isolates, a GWAS study on orthologs associated with this clonal complex was performed that identified several genes shown in Appendix A. In summary, 9 out of 10 CC241 isolates carried the plasmid p222 (99% identity and 97% coverage with the reference plasmid) encoding a bacteriocin with cytotoxic effect that was previously identified in a canine *S. pseudintermedius* isolate [16]. This p222 plasmid was also present in three (6%) non-CC241 isolates (two human (ST940 and ST946) and one canine (ST1356) isolate). The pRE25-like element that has been described as a chromosomal element carrying five resistance genes; *erm(B)*, *cat(pC221)*, *aph(3′)-III*, *ant6-Ia*, *sat4* and a toxin antitoxin system [15] was also associated with CC241 isolates. Sequence homology (>90%) with this element was detected in all CC241 isolates (*n* = 10) as well as in seven non-CC241 human isolates belonging to other sequence types belonging to ST940, ST946, ST989, ST1378, ST1380, ST1382, and in six non-CC241 canine isolates belonging to ST981, ST989, ST1351, ST1355, ST1356, and ST1361. Despite the sequence homology, all isolates missed one or more transposase genes (Appendix A). Manual investigation of the sequence assembly graphs using Bandage [20] revealed that collapsed repeats were the reason for these missing genes, a common issue with assembly from short-read sequences.

Next to the p222 plasmid and the pRE25-like element, a CRISPR-cas-type-III region was associated with CC241 and present in nine out of ten CC241 isolates. This CRISPR type was present as well in three other human isolates belonging to ST309, ST943 and ST946. All genes associated with either presence or absence in CC241 in the GWAS analysis are presented in Appendix A.

## 3. Discussion

Until a decade ago, human infections with *S. pseudintermedius* were only seldom reported, but the number of reports has markedly increased over the last decade, which could partially be linked to the use of MALDI-TOF [2,3,4,5]. Although zoonotic transmission from dogs to humans is generally suspected very few studies have compared the genetic characteristics of *S. pseudintermedius* isolates originating from human infections versus canine sources.

This study revealed that CC241 was overrepresented in the studied MSSP isolates and was composed mainly of human isolates. It is of interest to monitor the MDR CC241 clone as it carries genetic elements which can confer a number of selective advantages. First, the identification of a CRISPR-cas system associated with CC241 might increase genetic stability of CC241 which was shown for MRSP earlier [18], but not previously observed in MSSP. A second potential selective advantage could be the association of CC241 with plasmid p222, which contains BacSp222, a multifunctional peptide that functions as a bacteriocin against Gram-positive bacteria and is a virulence factor affecting eukaryotic cells [16]. A third potential selective advantage could be the multidrug resistance provided by the pRE25-like mobile element carrying five resistance genes coding for resistance to four antimicrobial classes [15]. This element with the IS1216 transposase on both ends is highly related to the pRE25 plasmid of *Enterococcus faecalis* and has recently been detected in canine isolates of *S. pseudintermedius* in Korea [15]. In the present study this pRE25-like element was present in all CC241 isolates, but was not limited to this clone, as it was present in 12 non-CC241 isolates as well. In the Korean study the STs of pRE25-like element containing isolates were diverse, but a selective advantage leading to dominance of clones containing this element was suggested [15]. To study in more detail the dissemination of MDR and the pRE25-like element we compared our isolate with two large studies with published canine MSSP genomes. The first one was in Europe where 8% of MSSP isolates were MDR and none had genes associated with pRE25-like (*erm(B)*, *cat_(pC221)_*, *aph(3′)-III*, *ant6-Ia*, *sat4*) [13]. The second in the USA showed 22.1% MDR among isolates containing *erm(B)*, *cat_(pC221)_*, *aph(3′)-III*, *ant6-Ia*, *sat4*, and 11 isolates carried the pRE25-like element [14]. Alignment with this element showed high homology with the pRE25-like elements identified in CC241 isolates in this study, and 4/11 isolates belonged to CC241 and the others belonged to various sequence types. The multiple findings of the pRE25-like element indicates that it is present in distinct MSSP lineages and most likely contributes to dissemination of resistance genes in MSSP, in the same way as has been observed for the resistance element carrying Tn5405 in MRSP [21]. The carriage of each of these elements will result in multidrug resistance and could limit treatment options for MSSP infections. The spread of this element among other human isolates could not be studied as none of the four publicly available MSSP genomes were multidrug resistant [12]. A similar finding of low phenotypic multidrug resistance (2 of the 24 tested isolates) has been reported for other human MSSP isolates [6].

Furthermore, we observed a high diversity in MSSP amongst both canine and human isolates, which confirms the diversity reported for canine MSSP isolates in a core genome MLST analysis [13]. To exclude that geographical differences are responsible for the observed diversity, a SNP phylogeny with publicly available MSSP genomes and our isolates was constructed. This confirmed the high diversity with small lineages, indicating that there is no geographical cluster of Dutch isolates, as they were dispersed over the tree. It also indicated that the lineage associated with CC241 was still overrepresented in human isolates [13,14] (Appendix A). This high diversity in MSSP is in contrast with the observed clonal dissemination of MRSP [13]. The phylogeny based on SNPs in the MSSP core genomes identified a deep-branched structure corresponding with a distinct MLST type for almost all isolates, whereas MRSP isolates mainly belong to a limited number of clonal complexes [13]. The genetic stability within certain MRSP clones might be explained by the carriage of lineage-specific prophages, restriction–modification or CRISPR/Cas systems hindering DNA uptake [18]. Furthermore, MRSP isolates often harbour resistance genes to multiple antimicrobial classes and use of any antimicrobial might co-select for the spread of these MDR isolates [21,22]. These elements might also explain the dominance of CC241 in our selection of MSSP isolates as all CC241 isolates were predicted MDR and contained lineage-specific elements, but were remarkably mainly found in human isolates (7/10). CC241 has also been described in a human infection in Spain [7], indicating a wider presence which is noticeable given the genetic diversity of MSSP.

Currently the number of *S. pseudintermedius* genomes from human infections is very low. This study provides the first genome comparison of human MSSP isolates, and analysis of more human isolates will be needed to unravel the importance of CC241 or other lineages in human MSSP infections.

GWAS analysis based on gene presence/absence showed no host association of specific genes which could be explained by the limited sample size but could also be due to the absence of host adaptation. The high sequence diversity of MSSP found in both sources might also play a role in the difficulty of finding host association. As an example, the genes encoding fibrinogen binding proteins (spsD and spsL) that were previously shown to be host associated [11,19] showed a high level of diversity, in both canine and human isolates (Appendix A). While these genes have been reported to contain repeat regions which might account for some diversity [23], in our isolates genetic variation was not limited to those regions. This variability in fibronectin binding protein encoding genes has not been previously reported. Studying allele variants rather than analysing gene presence/absence could be necessary to identify host-associated genes using, e.g., k-mer approaches. Several hundred isolates would probably be required to detect statistically significant associations because of the more severe multiple testing correction, as was shown when evaluating a k-mer-based approach in *Streptococcus pneumoniae* [24]. Analysis of protein variants or differences in protein domains, such as has been observed for leucocidins in *S. aureus* and *S. pseudintermedius* [25,26], was beyond the scope of this study.

No statistically significant difference was found between dog and human isolates in the carriage of resistance genes. Multidrug resistance was present in 41% of human isolates and 56% of canine isolates. The *tet(M)* gene was found slightly more often in dogs than in humans even though this was not statistically significant. Resistance genes, not located on pRE25-like elements, encoding for resistance against aminoglycosides (*aac(6′)-Ie-aph(2”)-Ia*), lincosamides (*Inu(A)*) and folate pathway inhibitors (*dfrG*), were only found in non-CC241 canine isolates. However, the sample size of human isolates was rather small. The prevalence of these genes was low in canine isolates which might explain why these genes were not identified in human isolates in this study.

## 4. Materials and Methods

### 4.1. Bacterial Isolates

In total, 57 MSSP isolates were included, 32 *S. pseudintermedius* isolates from human infections obtained from 6 Dutch hospitals between 2014 and January 2019, and 25 *S. pseudintermedius* isolates from canine infections isolated at the Veterinary Microbiological Diagnostic Centre were selected to match the years of isolation of human isolates. All isolates were selected based on convenience sampling and epidemiologically unrelated. Species identification was confirmed by matrix-assisted laser desorption ionization time-of-flight (MALDI-TOF MS) (Bruker MALDI Biotyper, Bruker Daltonics, Billerica, MA, USA).

### 4.2. Genome Analysis

DNA was isolated using the Qiagen UltraClean Microbial DNA isolation kit (Qiagen, venlo, the Netherlands. DNA libraries were prepared with the Illumina Nextera kit according to manufacturer’s instructions and sequenced using NextSeq sequencing with 150 base pairs reads (Illumina, San Diego, CA, USA). Reads-quality-check and adapter-trimming was performed with Trim Galore v0.4.4 (https://www.bioinformatics.babraham.ac.uk/projects/trim_galore/) Last update: 24 March 2017. The genomes were assembled with SPAdes v3.10.1 [27], and contigs smaller than 200 base pairs and with a coverage lower than 10 were removed. Genome quality was assessed with CheckM v1.1.2 [28] for completeness (>95%) and contamination (<5%). The genomes were annotated using Prokka v1.13 [29]. The batch upload function, including Resfinder and MLSTFinder, from the Center for Genomic Epidemiology (CGE) (Copenhagen, Denmark) [30] was used to analyse the resistance gene content and sequence types of all isolates (last accessed on 12 March 2019). In case of new sequence types, the alleles for Multi Locus Sequence Type (MLST) were assigned an ST number by the curator of the PubMLST database (https://pubmlst.org/spseudintermedius/) (accessed on 5 March 2019).

The whole genome sequences were aligned, and the core genome size determined, using Parsnp v1.2 [31] for phylogenetic single-nucleotide polymorphism (SNP) analysis of the core genome that was visualised using ITOL v4 [32] and a minimum spanning tree was made with Phyloviz 2.0 [33] using the goeBURST algorithm to assess the number of single-nucleotide polymorphisms between isolates. Orthology predictions of the annotated genomes were made using Roary 3.12.0 [34] and host- and clonal complex associated genes were determined using Scoary 1.6.16 [35] applying a threshold of *p* < 0.05 for statistical significance using Bonferroni correction. Identification of regions with significant genes was performed with a BLAST search against the NCBI database, resulting in the identification of two mobile elements (identity > 90% coverage > 80%). Mobile elements with Genbank accessions CP011490 and MK775653 were used as reference sequences for alignment to the MSSP sequence contigs using Geneious version 2020.1.1 (Biomatters, Auckland, New Zealand). Annotation of the reference element was used to predict the annotation of orthologous genes; if annotation in the reference was absent the annotation of Prokka was used. Statistically significant genes annotated as coding for hypothetical proteins by Prokka were further analysed using NCBI BLASTn 2.10.1. Geneious was also used for the alignment of *spsL* and *spsD* genes to their references in ED99 strain CP002478 and for construction of the *spsL* and *spsD* trees.

Pearson’s chi-square or Fisher’s exact test (when sample size <5) were used for statistical analysis for host association of resistance genes. A *p*-value of <0.05 was considered statistically significant with Bonferroni correction.

### 4.3. Data Availability

Whole genome sequence reads and assembled contigs have been deposited in the NCBI Sequence Read Archive under project number PRJEB39511, accession numbers are available in Appendix A.

## 5. Conclusions

In conclusion, canine and human *S. pseudintermedius* from the Netherlands were genetically highly diverse, and no host-specific genes could be detected. CC241 was overrepresented in human isolates. The CC241 isolates carried BacSp222, a bacteriocin with cytotoxic activity and the mobile genetic pRE25-like element carrying five resistance genes, which was more widely present in the MSSP population. The dissemination of the pRE25-like element could pose a threat for treatment of human and canine *S. pseudintermedius* infections.

## Figures and Tables

**Figure 1 antibiotics-10-00854-f001:**
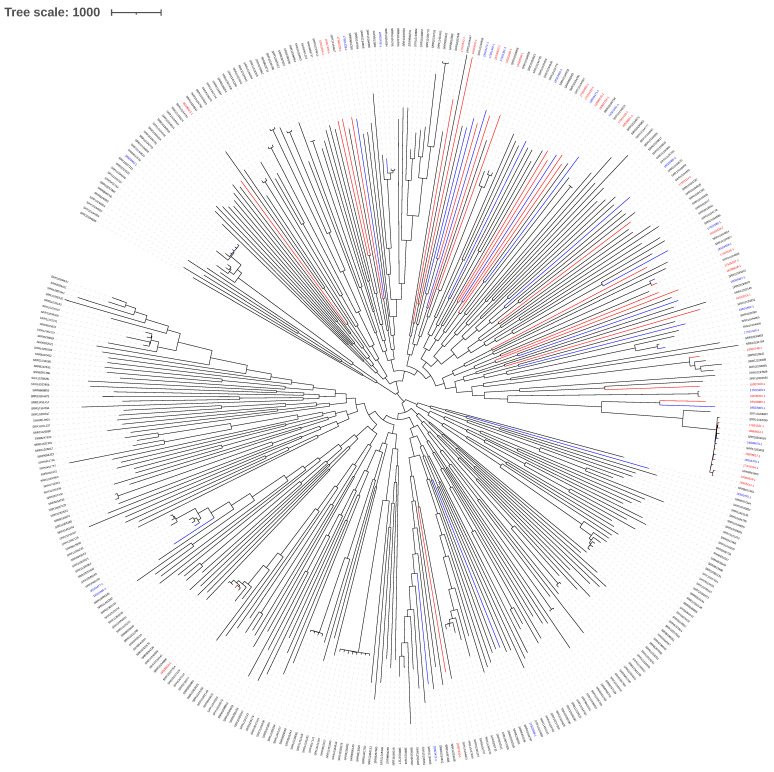
Phylogenetic tree based on the core genome SNPs of MSSP isolates. Publicly available MSSP genomes were compared with the MSSP isolates from this study, that have been marked red for human isolates and blue for dog isolates [12,13,14].

**Figure 2 antibiotics-10-00854-f002:**
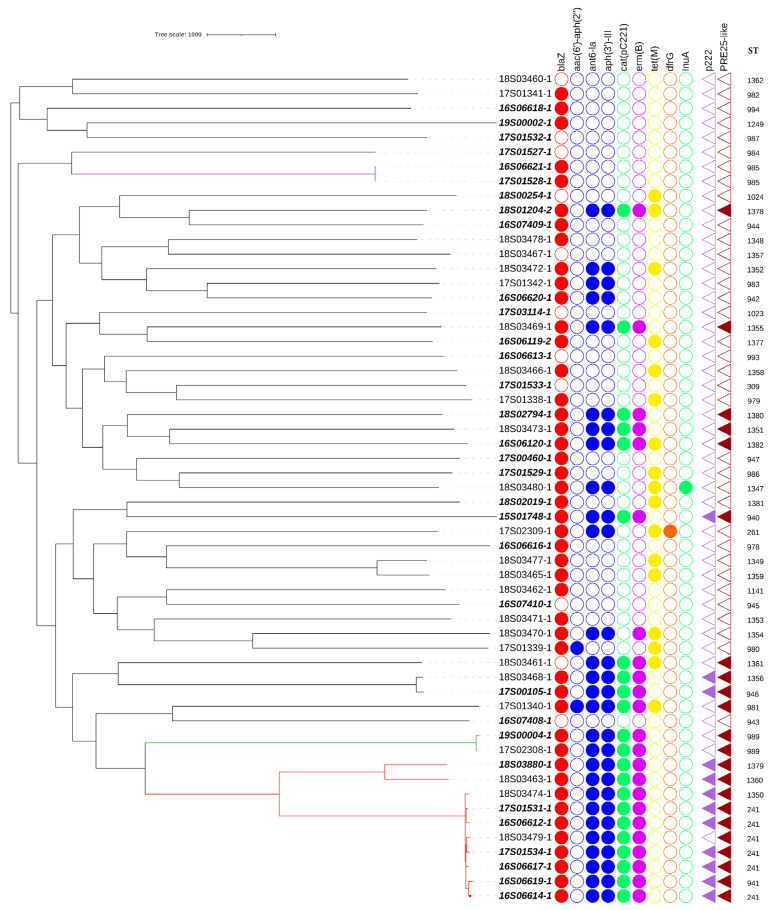
Isolate phylogeny and antimicrobial resistance genes.

**Table 1 antibiotics-10-00854-t001:** Antimicrobial resistance genes in human and canine *S. pseudintermedius* isolates.

Resistance	Gene	Human Isolates (*n* = 32)	Canine Isolates (*n* = 25)
β-lactam	*blaZ*	25 (78%)	23 (92%)
aminoglycoside	*ant6-Ia, aph(3′)-III*	14 (44%)	14 (56%)
	*aac(6′)-Ie-aph(2”)-Ia*	0	2 (8%)
chloramphenicol	*cat_(_* _pC221)_	13 (41%)	9 (36%)
macrolide	*erm(B)*	13 (41%)	10 (40%)
tetracycline	*tet(M)*	6 (19%)	11 (44%)
lincosamide	*Inu(A)*	0	1 (4%)
folate inhibitor	*dfrG*	0	1 (4%)

## Data Availability

Sequence reads are available under number PRJEB39511.

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
