# Peer review of "Absence of Host-Specific Genes in Canine and Human Staphylococcus pseudintermedius as Inferred from Comparative Genomics"

_antibiotics, 2021, doi:10.3390/antibiotics10070854_

Round 1
Reviewer 1 Report
Please, find the comments in the attached file

Author Response
Manuscript Title:
Absence of host specific genes in canine and human Staphylococcus pseudintermedius as inferred from comparative genomics
Journal: Antibiotics
Number: 1222514
Reviewer 1:
The authors describe an interesting data set to the study of methicillin-susceptible (MSSP) Staphylococcus pseudintermedius. There is little information in public repositories, and the authors describe almost half of the isolates both from humans and dogs to be multidrug-resistant (MDR), even when methicillin-susceptible and genetically diverse.
There are some concerns and a few suggestions and details that the authors can respond to without problem. The concerns are related to the lack of information about MSSP in the public repositories and the interest for other groups to ascertain the main differences of MSSP isolates when compared with methicillin-resistant ones.
We thank the reviewer for the constructive comments. We took all concerns and suggestions into account and have adapted the manuscript accordingly.
- As of the revision day, the accession numbers for either the SRA project PRJEB39511 and the different isolates in Suppl Table 1 are not available at NCBI. Please be sure the SRA PRJEB39511 project is accessible and the different accession numbers for the isolates in Suppl Table 1.
I would suggest the authors granted access to the deposited sequences for properly reviewing their results, confidentiality assured. It is imperative for us as reviewers to check the open-access of the sequencing data for the best benefit of our scientific community, even if it is under embargo in NCBI.
We checked the accession numbers of the project and the isolates, and all data is open access now. We have replaced read accession number with assembly accession number in Supplementary Table 1 to facilitate access to the assembly. Read files are available under SRA project PRJEB39511, which is described in lines 294-296
- The statement “data not shown” appears twice in the Results section.
118-119: We did not identify gene presence or absence significantly associated with either canine- or human S. pseudintermedius isolates using Roary (data not shown).
124-125: Alignment to reference genes spsD and spsL from ED99 showed a high level of sequence diversity across the gene with no clear clustering of variants of either dogs or human isolates (not shown).
Data not shown could be of outmost information for researchers in the field, especially for comparing gene content of the MSSP isolates described with other MSSP isolates from different geographical origins. Therefore, the suggestion will be to add it as supplementary information, and even if they were negative results, it would be helpful to other researchers in the field.
We have added the results of the Roary and Scoary analysis in Supplementary Table 2, and changed the current Supplementary Table 2 to Supplementary Table 3. The trees of spsL and spsD genes alignments have been added to the manuscript as Supplementary Figures 1 and 2. In the M&M section lines 288-289 we added; “Geneious was used to align spsL and spsD genes to their reference genes of strain ED99 (Accession number CP002478) and for construction of the spsL and spsD alignment trees.”
- Antimicrobial resistance genes are identified by WGS. However, no information is presented about the coincidence with the phenotype results of antibiograms. It seems to be in suppl table 1, with one of the tables for genotype and the other for the antibiogram, but it is not the case.
The presence of the resistance gene indicates AMR, but I’m not sure the authors can conclude the high frequency of MDR in MSSP without the corresponding phenotype confirmation by microbiology antibiogram. Therefore, the authors should clarify that the MDR consideration is based on the presence of the resistance gene.
Multiple studies have shown that resistance genotypes can accurately predict phenotypical antimicrobial resistance in Staphylococcus pseudintermedius (Wegener et al., 2018; Tyson et al., 2021). Therefore, the sentence “In S. pseudintermedius it has been shown that antimicrobial resistance genotypes can accurately predict phenotypical antimicrobial resistances (Wegener et al., 2018; Brooks et al., 2020; Tyson et al., 2021)” was added lines 101-102. To point out that the MDR consideration is based on the presence of the resistance genes, the word “predicted” was added before MDR in line 28 (replacing detected) in lines 29, 30,108 and 213.
- Related to that point, authors could compare MDR frequency in other MSSP genomes in NCBI or ENA. There are some of them publicly available, and it would be interesting to ascertain if MDR is also associated with a plasmid, as it seems to be the case here.
We agree with the reviewer that comparison of the MDR distribution in a larger set of MSSP genomes could provide broader insight in the epidemiology of MDR isolates. We have compared our results with published canine genome collections, where multidrug resistance was analysed (Tyson et al, 2021 and Haenni et al., 2020). Unfortunately, human MSSP genomes were restricted to only 4 genomes published by Little et al, 2019. This analysis was added in the discussion of the manuscript in lines 180-196. We have discussed the MDR proportion in published genomes, indicating the 22% MDR in a collection of canine MSSP isolates from the USA of which 11 isolates carried the erm(B), cat(pC221), aph(3’)-III, ant6-Ia, sat4 genes and similar pRE25-like elements, as identified in our study, were detected (Tyson et al., 2021). Furthermore, a lower MDR prevalence was observed in an European canine MSSP collection of isolates where the proportion MDR was 8% (Haenni et al., 2020). None of the four MSSP genomes from human isolates were multidrug resistant, and a low (2/24 isolates) have been reported to display phenotypic multidrug resistance in human MSSP isolates (Somayaji et al., 2016).
- Please indicate the average nucleotide identity among isolates, e.g., by running FastANI (Jain et al, 2018) and provide (1) average values and ranges in the text for human- and dog isolates, and (2) a suppl table with the ANI matrix and shared regions as provided by the FastANI output.
We ran the FastANI analysis as suggested by the reviewer. The average nucleotide identity (ANI) of the isolates ranges between 98.9% and 100.0%. Human-human ANI ranges from 99.0-100.0%, dog-dog ANI ranges from 99.1-99.9% and dog-human 98.9-100.0%. All with a mean of 99.3%. The resolution of this technique does not allow to draw significant conclusions.
In our opinion this analysis does not add relevant information compared with the phylogenetic analysis. FastANI is a tool used for species delineation (Thompson et al., 2013) and in our study we focused on a single species only, S. pseudintermedius. Therefore, we chose not to add the results of the FastANI analysis to the manuscript.
- Please, indicate the size of each one of the genomes in supplemental table 1 and how the average core genome size of 2,170,170 bp was calculated in M&M section.
The genome sizes have been added to Supplementary Table 1.
In the M&M section the sentence was changed to indicate how the core genome size was determined “The whole genome sequences were aligned, and the core genome size determined, using Parsnp v1.2” (line 272).
- Methicillin susceptible?? Not in suppl table 1. Please clarify Suppl Table 1 (two tables with almost the same columns)
We apologize for the mistake of adding two similar tables in the file. The additional table has been removed, and MSSP has been added in the title of the table.
Please add p222 & PRE25-like definitions in the Fig legend
This has been changed in the figure legend to “presence of mobile genetic element PRE25-like (Kang and Hwang, 2020) and plasmid P222 (Wladyka et al., 2015)(filled triangles).
Please add comprehensive legends to suppl tables
All legends of the supplementary tables have been adapted.
Change “resistences” to “resistances” in table1 and +suppl table 1
This has been changed.

Reviewer 2 Report
To the editor and the authors:
The research question is important. Staphylococcus pseudintermedius is an important pathogen in dogs and associated with infections in humans as an opportunistic pathogen. In addition, there are few studies about this bacterial species in the scientific literature. So it is necessary to identify the main Staphylococcus pseudintermedius genetic lineages currently disseminated in human and animal infections. It is also important to characterize antibiotic resistance, virulence and genes associated to these phenotypes.
The whole manuscript describes a scientific study with a good sampling (57 bacterial isolates, including 32 from human and 25 from dog infections). The methodological steps seem Ok and well described (although they are very summarized). It is meaningful to sequence whole-genomes and the authors seem to have obtained very interesting results.
However the authors do not present very well the results. It seems they have very interesting bacteriological and genomics results that could be much better explored. With new and more complete analyses, the discussion could be improved. The authors should also review other important and recent scientific articles. The scientific article would be at a higher level.
Some specific suggestions are pointed below:
- The epidemiological aspects could be better described and explored in the text (mainly in the topic 2.1). The authors should describe the geographical locations of the hospitals where the samples were collected and the period of the study, for example. This could be later compared with the lineages / sequence types identified in the study.
- The bacteriological characteristics of the isolates were not presented. Besides MALDI-TOF, what are the other bacteriological results? Is there any information about the phenotypic antimicrobial resistance of all isolates?
- The sequence types and clonal complexes detected should also be described in more detail. Some STs from only one clonal complex was very emphasized (cc241), while the others STs are totally forgotten (they are only presented in the Supplemental Table 1).
- The phylogenetic characterization is also poor. At the present time, with whole-genomes more easily sequenced, it would be important to carry out a more complete genetic analysis of the isolates comparing with other reference Staphylococcus pseudintermedius isolates. A more complete phylogenetic tree would help to demonstrate the comparison with other genomes and to discuss better the overall results.

Author Response
Manuscript Title:
Absence of host specific genes in canine and human Staphylococcus pseudintermedius as inferred from comparative genomics
Journal: Antibiotics
Number: 1222514
Reviewer 2:
To the editor and the authors:
The research question is important. Staphylococcus pseudintermedius is an important pathogen in dogs and associated with infections in humans as an opportunistic pathogen. In addition, there are few studies about this bacterial species in the scientific literature. So it is necessary to identify the main S. pseudintermedius genetic lineages currently disseminated in human and animal infections. It is also important to characterize antibiotic resistance, virulence and genes associated to these phenotypes.
The whole manuscript describes a scientific study with a good sampling (57 bacterial isolates, including 32 from human and 25 from dog infections). The methodological steps seem Ok and well described (although they are very summarized). It is meaningful to sequence whole-genomes and the authors seem to have obtained very interesting results.
However the authors do not present very well the results. It seems they have very interesting bacteriological and genomics results that could be much better explored. With new and more complete analyses, the discussion could be improved. The authors should also review other important and recent scientific articles. The scientific article would be at a higher level.
We thank the reviewer for the constructive comments and for the suggestions to improve the manuscript. We took all concerns and suggestions into account and have adapted the manuscript accordingly.
- The epidemiological aspects could be better described and explored in the text (mainly in the topic 2.1). The authors should describe the geographical locations of the hospitals where the samples were collected and the period of the study, for example. This could be later compared with the lineages / sequence types identified in the study.
We agree that geographical information on the hospitals is informative when studying the epidemiology and have added the sentence “Patient isolates were obtained from hospitals located in 6 different provinces in the Netherlands.” (line 70). The hospital location was added in Supplementary Table 1. Also, information on collection period was added in the sentence “All isolates were obtained between 2014 and January 2019.” And “Isolates were isolated from individual patients at different time points and were to the best of our knowledge epidemiologically unrelated” was added in lines 74-76. In the materials and methods the sentence about isolate selection was changed into: “All isolates were selected based on convenience sampling and their epidemiological unrelatedness” (line 253).
- The bacteriological characteristics of the isolates were not presented. Besides MALDI-TOF, what are the other bacteriological results? Is there any information about the phenotypic antimicrobial resistance of all isolates?
MALDI-TOF was used to ensure species identification as it is a reliable tool, in particular as we use an additional in-house database facilitating proper identification of S. pseudintermedius and distinction of other staphylococci within the SIG group. No antimicrobial susceptibility testing was performed on the isolates. Previous studies have shown that resistance genotypes accurately predict phenotypical antimicrobial resistance in Staphylococcus pseudintermedius (Wegener et al., 2018; Brooks et al., 2020; Tyson et al., 2021). Therefore, the sentence “In S. pseudintermedius it has been shown that antimicrobial resistance genotypes can accurately predict phenotypical antimicrobial resistances (Wegener et al., 2018; Brooks et al., 2020; Tyson et al., 2021).” was added to lines 101-102. To point out that the MDR consideration is based on the presence of the resistance gene, the word “predicted” was added before MDR in lines 28 (replacing detected), 29, 30,108 and 213.
- The sequence types and clonal complexes detected should also be described in more detail. Some STs from only one clonal complex was very emphasized (cc241), while the others STs are totally forgotten (they are only presented in the Supplemental Table 1).
We agree that we did not mention the other STs. As the genotypes of MSSP were highly diverse and most sequence types were only represented by a single isolate, we have added “In total the isolates belonged to 50 different sequence types (ST)”in line 80 and have added that they represented single isolates in line 90.
- The phylogenetic characterization is also poor. At the present time, with whole-genomes more easily sequenced, it would be important to carry out a more complete genetic analysis of the isolates comparing with other reference S. pseudintermedius isolates. A more complete phylogenetic tree would help to demonstrate the comparison with other genomes and to discuss better the overall results.
Given the number of publicly available genomes a phylogenetic comparison is tempting, but the outcome of the analysis is influenced by the high geographical diversity, biased source distribution and the genotypic diversity of MSSP. For identification of potential host-associated genes and clones it is necessary to limit the variables (geographic, time) and we therefore specifically included isolates that were obtained in the same time period and the same geographical location. However, we did perform a new phylogenetic analysis with publicly available genomes as this comparison could provide other insights. Therefore, we constructed a database with published and publicly available genomes. Two large MSSP collections of well characterized and published genomes and the reference genome ED99 as well as the 4 available human genomes, were included (Little et al., 2019; Haenni et al., 2020; Tyson et al., 2021)
After assembly and quality control, a set of 444 genomes (from NCBI 383 from canine and 4 from human isolates) with 57 genomes from our study were included. The phylogenetic analysis has been added in Suppl. Figure 3 and is being discussed at lines 198-203 of the discussion.
The limited number of human genomes was discussed in lines 217-220
Therefore my main suggestion to the authors is to revise the whole manuscript (major revision) to describe and to present the results in more detail. After, it is also necessary to improve the Discussion.
As suggested by the reviewer we have provided more details in the result and have revised the discussion.
Round 2
Reviewer 2 Report
The authors responded very well to my main comments about the article. They also performed the main suggested modifications. The presentation of the Results and the Discussion sections are much better now. Also recent scientific references were included. The whole manuscript has been really improved.
However I still suggest the authors to revise some minor aspects in the manuscript text. First, the supplementary figure 3 could be inserted in the article as the first Figure aiming to present the overall diversity of the isolates. In addition, I would recommend highlight the sequences of your study (preferentially with different colors to isolates from human and canine patients sources). Second, the authors should standardize the whole text. Only one example in the topic “MSSP-infections determinants”: “Other infections were ear (n=2), 67 joint (n=2), skin (n=2) , systemic (n=1), urinary tract (n=1) and rectum infections(n=1).” “The majority of canine isolates were from ear infections (n=9/25), followed by skin infection (n=8/25), wound infections (n=5/25), urinary tract infections (n=2/25) and a joint infection (n=1/25).” You can choose (n=X or n=X/Y), but please use the same style in the whole text. A detailed revision focusing in text standardization would be welcome in the whole manuscript.
With these minor modifications, the article could be accepted. It is not necessary a new revision.
Author Response
We thank the reviewer for the constructive comments. We have adapted the manuscript accordingly.
The figure with the complete tree has been added as figure 1 as suggested, the isolates of dogs and humans from our study have been marked in blue and red respectively. As the figure is very large it is difficult to read without zooming in, which is why this was originally added as supplementary material.
We have added Figure 1 also in a separate pdf file.
The text of the manuscript has been standardised as suggested.